

# Environmental controls on marine productivity

# near Cape St Francis, South Africa

Mark R Jury

University of Zululand, KwaDlangezwa, 3886, South Africa

and Physics Dept. Univ. of Puerto Rico, Mayaguez, USA, 00681

## Abstract

This study considers ocean-atmosphere influences on marine productivity over the shelf near Cape St Francis, South Africa. Multi-day estimates of chlorophyll fluorescence in the period 2006-2017 with an area: 34.5-33.75S, 24-26.5E, provide the basis for evaluation using data from high resolution reanalysis.

Correlations with the mean annual cycle of chlorophyll fluorescence were significant for salinity, linking marine productivity and the coastal hydrology. A strengthened Agulhas Current induces cyclonic shear that lifts water at the shelf edge. Composite high chlorophyll fluorescence events were dominated by a large-scale mid-latitude atmospheric ridge of high pressure. The resultant easterly winds caused offshore transport and the upwelling of cool nutrient-rich water, in multi-day events at the beginning and end of austral summer. Environmental controls on inter-annual fluctuations of the commercial fishery were also explored. Southwestward currents and diminished heat fluxes favoured squid catch, while anchovy and sardine were linked with upper northerly wind, consistent with large-scale weather patterns that underpin coastal upwelling and river discharge. Productivity lags a few days behind cyclonic wind and current shear and the upstream coastal hydrology, which shares a common atmospheric driver.

mark.jury@upr.edu



## 1. Introduction

The southern coast of South Africa is swept by the prevailing warm Agulhas Current (Lutjeharms et al. 2000). Shelf-edge upwelling is induced by cyclonic shear of the current and downstream widening of the shelf (Schumann 1986, 1988, Lutjeharms 2006, Goschen et al. 2015, Malan et al. 2018). Coastal winds average 7 m/s (Schumann and Martin 1991) and tend to blow from east in summer and from west in winter. The easterlies lift cold nutrient-rich water next to three Capes: Padrone, Recife, St Francis, separating two sheltered bays (Schumann et al. 1982, Goschen and Schumann 1987, Schumann 1999, Schumann et al. 2005, Roberts 2010, Goschen and Schumann 2011, Pattrick et al. 2013). Rivers of the Eastern Cape discharge fresh silty water up to 100 m$^3$/s, during infrequent floods in the Gamtoos, Sundays and Fish River Valleys (Bladeren et al. 2007). Their plumes spread along the shore and into the large bays (Scharler and Baird 2005) and promote stratification in summer. Intra-seasonal fluctuations are attributed to pulsing and meandering of the Agulhas Current (Lutjeharms et al. 1989, Lutjeharms and Roberts 1988, Goschen and Schumann 1990, Rouault and Penven 2011) and by coastal low pressure cells coupled to shelf waves that pass eastward around South Africa, bringing changes in marine weather (Jury et al. 1990, Schumann and Brink 1990). Air-sea interactions exhibit cross-shore gradients that are important for coastal resources (Beckley 1988). The inshore bays are sheltered and develop stable layers during summer that aggregate phytoplankton. In this area known as the eastern Agulhas Bank, a commercial fishery aimed at sardines (67 K T/yr), squid (3K T/yr), anchovy, roman, mackerel, etc (Roberts et al. 2012, Cochrane et al. 2014) sees most catch effort within 50 km of the coast surrounding St Francis Bay (34S, 25E), which offers protection from persistent SSW swells (avg 2.3 m).

This study considers marine productivity at multi-day time-scales off Cape St Francis, South Africa. Using high resolution model-assimilated observations, our analysis seeks to understand how environmental conditions modulate chlorophyll and salinity. Section 2 covers the data and methods, while Section 3 presents the results progressing from mean features and statistics to case studies of productive events. Context is provided by temporal cross-correlations, spatial point-to-field regression and statistical comparison of commercial fish catch with environmental variables. Section 4 provides a discussion that relates the bio-chemistry indices to ocean-atmosphere forcing.



## 2. Data and Methods


A study is delimited to the productive eastern Agulhas Bank in the period of high
resolution satellite coverage. Ocean conditions off Cape St Francis, South Africa (32.5–35S,
22–28.5E, Fig 1a) were described using multi-day SODA3 (25 km horizontal resolution,
Carton et al. 2018) and daily HYCOM (GOFS3.1) reanalysis (9 km, Chassignet et al. 2009).
These reanalysis (cf. Appendix) assimilate insitu observations from ship, buoy and drifter,
and satellite infrared and microwave radiometer measurements over a multi-day period to
produce a hindcast with a vertical resolution of < 5 m near the surface. Local validations are
reported in Jury and Goschen (2019). Sea surface temperature (SST) was assimilated via
daily 1 km infrared and 9 km microwave data after de-clouding and calibration (Chin et al.
2017). Salinity was assimilated via wide-swath passive microwave radiometers, while
multiple zenith-pointing radar altimeters determined sea surface height and near-surface
currents. Chlorophyll was estimated from MODIS satellite 4 km resolution green-band data,
atmospheric corrected using an 8-day composite maximum and adjusted for radiometer drift
(Hu et al. 2012). Similarly, the fluorescence line height was estimated from level-3 MODIS
red-band data, which is less sensitive to coastal sediments (Gower 2016, Houskeeper and
Kudela 2017). Swell characteristics were derived from the Wave-watch v3 hindcast (Tolman
2002). Atmospheric conditions were described using daily MERRA v2 reanalysis (50 km
resolution, Gelaro et al. 2017). Winds were assimilated from station, ship and buoy
measurements, blended with satellite cloud drift and active microwave scatterometer
retrievals. The ocean reanalyses are embedded within the coupled data assimilation system:
HYCOM from NCODA (Cummings and Smedstad 2013), and SODA3 from MERRA2 that
incorporates hydrology and air chemistry (Reichle et al. 2017).
Daily and 8-day temporal records were extracted for an index-area 33.75–34.5S, 24–
26.5E encompassing the shelf around Cape St Francis, South Africa (cf. Fig 1a). With two
large bays, a widening shelf, and the offshore Agulhas Current, the index area is subject to a
variety of processes. The analysis was confined to the period 2006–2017, when high
resolution satellite coverage of the shelf zone facilitates analysis of SST, winds and currents.
The coastal hydrology was described by CHIRPS v2 rainfall (5 km resolution; Funk et al
2015), CMORPH satellite rainfall (25 km; Joyce et al. 2004) and SADW discharge records
for the Gamtoos, Sundays and Fish Rivers. The marine variables include: SST, sea level air
pressure (SLP), winds, heat and radiation fluxes, salinity, currents, vertical motion (cf. Table
1). An 8-day resolution compatible with MODIS chlorophyll fluorescence yields a record
length of 552 comprised of 46 x 12 years (1 Jan 2006 - 31 Dec 2017). From this, multi-day



events were identified by ranking the index-area chlorophyll fluorescence and salinity at 10
m depth. The chlorophyll fluorescence time series was subjected to wavelet spectral analysis
to determine the degree of cyclicity and its amplitude and period.
Statistical associations were studied by pair-wise correlation of the mean annual cycle
of chlorophyll fluorescence and 10 m salinity with a variety of environmental parameters
listed in Table 1. Statistical significance above 90% confidence was achieved with the
Pearson-product moment r > |0.62| for 6 degrees of freedom. Relationships with chlorophyll
fluorescence motivated a point-to-field correlation analysis of the salinity record and large
scale fields of Oct-Mar sea level air pressure, covering 25–45S, 10–43E, 2006-2017. A
composite average of the top-10 chlorophyll fluorescence events was conducted to study
anomaly patterns for 500 hPa geopotential height, satellite rainfall and surface zonal winds.
Composite maps were analyzed sequentially at -4, -3, -2, -1, 0 days before the time of peak
ocean colour, to reveal the large scale atmospheric forcing. The top-10 events began on the
following dates: 1 Nov 2009, 1 Nov 2014, 24 Oct 2014, 24 Oct 2009, 17 May 2015, 19 Dec
2007, 9 Nov 2014, 23 Oct 2012, 10 Feb 2011, 23 Apr 2015, ranked by value. Exploratory
analysis of Agulhas Current pulses (cf. Appendix Fig A2), were made by calculating
HYCOM daily sea surface height variance 2006-2017 per 0.1° longitude bin from 24-29E on
34.5S, and via case study hovmoller sequences across the shelf.
Inter-annual fluctuations of commercial fisheries and environmental conditions were
explored using monthly ocean reanalysis appropriate for longer time scales: Jan-Jun season
SODA v3 ocean reanalysis (Carton et al. 2018), satellite SST (Chin et al. 2017), and annual
catch data from local and international sources (van Zyl and Willemse 2000, J. Coetzee pers.
comm. 2018). Pair-wise correlations were calculated for the main fishery species in the
period 1981-2015; 90% confidence is reached with r > |0.40| for 35 degrees of freedom. The
websites used for data extraction and analysis are listed in the acknowledgements.
**3. Results**
**3.1 Marine climate and annual cycle**
Mean maps of the study area topography and SST are illustrated in Fig 1a,b. Sharp
gradients in the mean SST field were evident: inshore waters are 17-19C while offshore
waters are 22-24C. Shelf-edge upwelling next to the warm Agulhas Current divided the two
regimes (Lutjeharms et al. 2000, Malan et al. 2018). Over the cool inshore waters, the
sensible heat flux was low (Fig 1c) and the atmospheric boundary layer was shallow (800 m).
So the coastal topography steers the wind to a longshore axis (Schumann and Martin 1991)



that further sharpens the gradients.

Figure 1d-f illustrates the mean annual cycle of key variables. Chlorophyll

fluorescence showed bi-modal peaks in Oct-Nov and Mar-Apr, of importance to the marine
food web and fishery abundance (Shannon et al. 1984, Hutchings 1994). The wind stress curl
reached a cyclonic condition from Oct-Mar that favoured upwelling via easterly wind shear
attending sub-tropical high pressures. During winter season, anticyclonic curl via westerly
wind shear promoted downwelling. The annual cycle of 10 m salinity revealed a summer
minimum, induced by rainfall and river discharge from November to March. A histogram of
daily zonal winds (Fig 1g) showed near equal occurrence of upwelling (-U) and downwelling
favourable conditions, and many instances of 5-10 m/s in both east and west sectors. A case
study upwelling event with high chlorophyll fluorescence 24 Oct - 8 Nov 2009 is illustrated
in the appendix (Fig A1), and links easterly winds, rainfall and marine productivity.

## 3.2 Chlorophyll and water flux

The mean MODIS chlorophyll and fluorescence maps (Fig 2a,b) show higher values

inshore and around the capes, and lower values outside of the Agulhas Current. The
chlorophyll tended to hug the coastal zone and its wave- and river- suspended sediments. In
contrast the fluorescence exhibited an offshore plume from Cape Padrone, associated with
shelf-edge upwelling. The chlorophyll fluorescence index-area time series (Fig 2c) is
punctuated by spikes of 10 mg/m$^3$, which identify key events mainly at the beginning and end
of summer. Its wavelet spectral energy (Fig 2d) exhibited fluctuations from 20 to 45 days in
early and late summer bursts. The years 2009-2011 experienced greater 20-50 day pulsing,
while 80-90 day oscillations of chlorophyll fluorescence characterized the years 2014-2015
and 2017. At 8-day resolution, higher frequencies were unresolved.

The mean HYCOM water flux into the ocean (Fig 3a) exhibited a low axis along the

shelf edge. Marine rainfall was a feature outside and east of the Agulhas Current. Significant
river discharges on the upstream coast and in the bays were measured (Fig 3b). The Gamtoos-
Sundays River output was 1-10 m$^3$/s; while the Fish River reached ~ 100 m$^3$/s during wet
spells. Yet there were prolonged dry spells characterized by minimal discharge. The case
study section considers seasonal changes from 2010 to 2011 and a detailed analysis of an
event in 2014.

## 3.3 Case study productivity and salinity events

The river discharge time series reflects a period of change from dry to wet conditions

from Aug 2010 to Mar 2011. Rainfall maps for those contrasting periods (Fig 3c,d) exhibited





dry conditions over the Eastern Cape followed by widespread rainfall. River discharges rose
from 1 m$^3$/s to ~100 m$^3$/s; and the shelf exhibited increased chlorophyll (Fig 3e,f). The main
axis of the Agulhas Current was inshore early in the event and shifted seaward later in the
event. Initially there was northeastward flow along the coast that diminished later.

Figure 4a-g follows the development of a low salinity event from 15-30 Oct 2014.

Initially there was salty water over the shelf, but gradually the upstream rainfall and river
discharges (~35 m$^3$/s) fed southwestward into a buoyant plume. By the end of Oct 2014, the
near-surface salinity over the shelf declined to 35.0 ppt. The daily 10 m salinity record
reveals this event to be the lowest in a decade. The large-scale wind map shows SE flow over
South Africa, driven onshore by a trough in the Mozambique Channel and a mid-latitude high
pressure cell (Fig 4e). Subsequently, there was a noteworthy increase in chlorophyll from
mid-October to early November 2014 (Fig 4f,g). An increase in upstream rainfall and run-off
promoted water turbidity and marine productivity.

Vertical sections of zonal wind, currents and sea temperature are shown in Fig 5a-c

averaged over the high chlorophyll event (17 Oct - 8 Nov 2014). Along the 26E longitude
these sections identify the easterly low level jet over the shelf and associated cyclonic wind
shear. The upwelling-favourable easterly winds were vertically capped at 850 hPa (1.5 km).
The wind stress vorticity was ~ 10$^{-6}$ N m$^{-3}$, lifting water near the coast. The vertical section of
zonal currents revealed a westward Agulhas Current of 1.4 m/s between 34.7-34.3S latitude.
Near the coast zonal currents were zero, hence the cyclonic vorticity at 34.2S was ~ 3 10$^{-5}$ s$^{-1}$,
lifting water at the shelf edge. The sea temperature section exhibited warm 21C sea
temperatures offshore, typical of the Agulhas Current during spring. Inshore there was
pronounced upwelling of 12C water lifted from ~ 80 m depth at 34.1S, breeching the surface
with a temperature of 14C.

Vertical motion is generated by vorticity of the longshore wind stress and current

(Hsueh and O'Brien 1971, Gill and Schumann 1979, Blanton et al. 1981, Brink 1998)
according to: $\zeta_\tau / \rho\, f = W = \zeta_{U10} (C_D\, Z / f\, dt)$. Using the above vorticity values, water density
($\rho$) 10$^3$ kg m$^{-3}$, coriolis (f) 7.7 10$^{-5}$ s$^{-1}$, bottom drag ($C_D$) 10$^{-2}$ (Liu and Gan 2015), depth of
current shear (Z) 2 10$^2$ m, over a multi-day time (dt) 10$^5$ s. The wind stress curl and the
current shear each generate vertical motion of ~ 10$^{-5}$ m/s. In the 2014 case study, the easterly
winds and currents combined to lift water off Cape Padrone. Yet half of the time winds are
from the west (cf. Fig 1g) and oppose the current-induced upwelling.

An exploratory analyses of processes underpinning shelf-edge upwelling was made.

Appendix Fig A2 illustrates that variance of sea surface height is greatest outside the Agulhas



Current. Hovmoller plots of sea surface height during peak chlorophyll fluorescence events
exhibited gradual strengthening of the cross-shelf gradient, as  pulses of +SSH moved
westward at ~0.2 m/s outside the Agulhas Current and wind-driven offshore Ekman transport
induced −SSH inshore.

### 3.4 Statistical insights

The statistical analysis of temporal records reveals a link between coastal rainfall > 10
mm/day and chlorophyll (Fig 6a) particularly in summer (r > 0.6) when run-off is greater.
According to the point-to-field correlation map (Fig 6b), reduced salinity during summer is
associated with a low pressure trough over the Benguela and a high pressure in the mid-
latitudes southeast of Africa, which together promote upwelling favourable easterly winds.
The correlation map with respect to sea surface height anomalies (Fig 6c), shows that reduced
salinity is linked with a low-inshore / high-offshore gradient, hence accelerated shelf-edge
currents and shear-induced upwelling.
Figure 7 illustrates a sequence of composite 500 hPa geopotential height anomalies
for the top 10 chlorophyll fluorescence events. The maps follow the eastward movement of a
mid-latitude ridge, from 4 days before, to the day of maximum colour in the index area. The
ridge moved 10° longitude per day on 50S, and generated easterly wind anomalies of 10 m/s
on the shelf edge and heavy rainfall over the Eastern Cape interior. Hence large-scale weather
conditions promoted marine productivity through the concurrence of upwelling and river
discharge.
Table 1 lists the pair-wise cross-correlation between the mean annual cycle of
chlorophyll, salinity and other variables from the index area. Chlorophyll fluorescence is
most correlated with salinity (r = -0.62) followed by meridional and zonal current (-0.56).
Processes that enhance chlorophyll fluorescence over the annual cycle are water mass
freshening and an intensified Agulhas Current (cf. Fig 6c, Fig A2). For salinity, annual cycle
correlations were generally higher. The strongest relationships were with sea level air
pressure (r = 0.91), net longwave radiation (0.91) and wave height (0.90), indicating that
lower air pressure, less outgoing radiation (eg. greater cloudiness) and smaller wave height
coincided with lower salinity over the annual cycle.
Annual fish catch data (Fig 8) provide a basis to evaluate environmental influences, as
listed in Table 2. Year-to-year changes of squid, anchovy and sardine catch displayed weak
relationships with many marine variables. Only squid catch exhibited inter-annual variability;
anchovy catch was minuscule and sardine catch showed decadal oscillations. Negative





(southwestward) currents and diminished heat fluxes favoured squid catch. Anchovy and
sardine catch increased following a season with anomalous upper northerly wind (V 2, r = -
0.37), consistent with the large-scale summer-time weather patterns in Fig 6c that underpin
increased productivity (cf. Jury 2011, Fig 5 therein).

## 4. Discussion and conclusion

The relative role of local and remote atmosphere and ocean forcing on marine
productivity over the shelf near Cape St Francis has been explored. The 8-day MODIS
chlorophyll fluorescence and HYCOM salinity in the period 2006-2017 were the primary
descriptors in an index-area 34.5-33.75S, 24-26.5E. A variety of atmospheric and oceanic
variables were obtained from high resolution reanalysis products such as HYCOM, and
annual cycle correlations were explored.
It was recognized that ocean colour, as a proxy for marine productivity, has
uncertainties due to potential contamination by clouds and aerosols, and by coastal wave- and
river- induced sediments. This uncertainty was addressed by the addition of red-band
fluorescence line height to the traditional green-band chlorophyll concentration. Over the
annual cycle, salinity and ocean currents exhibited significant negative relationships with
chlorophyll fluorescence. Furthermore, it was also recognized that the ocean reanalysis are
based on coupled data assimilation that benefits from satellite technology but limited insitu
calibration. Although the work infers that the ocean reanalysis is equivalent to reality, there is
uncertainty that limits understanding. This does not prevent explorational studies as reported
here, but could inhibit translating outcomes into strategic decisions.
The mean pattern of chlorophyll fluorescence revealed an axis of high values off Cape
Padrone, where current-edge upwelling is prevalent (Swart and Largier 1987, Lutjeharms et
al. 2000, Lutjeharms 2006 pp.140-146). Downstream widening of the shelf bathymetry
elongates this cold tongue toward the wind-driven upwelling plumes off Cape Recife and
Cape St Francis. This shelf-edge feature modulates the location of stable layers and primary
production. From October to March,  a strong vertical temperature gradient tends to
concentrate phytoplankton at depths > 30 m (Probyn et al. 1994), at the bottom of the
euphotic zone and wind mixed layer.
The annual cycle analysis presented here revealed that marine productivity peaked in
early and late summer when sub-tropical cut-off lows were most frequent (Favre et al. 2013).
The large-scale point-to-field analysis indicated that a mid-latitude high pressure ridge
underpins marine productivity, reducing salinity via upstream rainfall / coastal run-off and
promoting wind-driven coastal upwelling and cross-shelf SSH gradients that accelerate shelf-



edge currents and upwelling. Case studies and the top-10 composite revealed similar features
in chlorophyll fluorescence events: they follow a spell of sustained easterly wind-driven
coastal upwelling and low salinity induced by local and upstream rainfall and river discharge.
The competing influences of: i) coastal run-off ~100 m³/s, ii) marine rainfall, iii) air-sea
interactions, iv) inshore upwelling ~10 m/day, and v) intrusions from offshore; can not be
resolved in this exploratory work and deserve further study. Yet the evidence during
productivity events indicates that cyclonic shear of easterly winds and shelf-edge currents (cf.
Fig 5a,b, Fig 6c, Fig A2) play prominent roles to lift water and generate high chlorophyll
fluorescence along the coast. Statistically, about half the time westerly winds oppose the
currents and suppress marine productivity, mainly during winter.
Earlier findings on multi-day upwelling events (Goschen et al. 2012) appear
complimentary to those reported here for co-varying indices of marine productivity. High
chlorophyll fluorescence lags a few days behind cyclonic wind and current shear and the
upstream coastal hydrology, which shares a common atmospheric driver. Environmental
controls on inter-annual fluctuations of the commercial fishery were explored using ocean
reanalysis appropriate for longer time scales (monthly SODA3). Southwestward currents and
diminished heat fluxes favoured squid catch, while anchovy and sardine were linked with
upper northerly wind, and large-scale weather patterns that underpin coastal upwelling and
river discharge (cf. Jury 2011). Earlier work determined physiographic preferences for
pelagic fish on the East Agulhas Bank (Armstrong et al. 1991), some of which are reflected in
the results of Table 2 here. Further work will analyze the pulsing of the Agulhas Current and
its affect on environmental conditions over the shelf from intra-seasonal to multi-year time
scales.

## Acknowledgements

Most environmental data were sourced from websites of the International Research
Institute for Climate, Climate Explorer of the Netherlands Meteorological Institute, NASA
Giovanni, and University of Hawaii Asia-Pacific Data Resource Center. Fish catch derived
from Fishbase, supplemented by South African government sources. River discharge data
were obtained from the South African Dept of Water, Hydrology Services website.

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



**Table 1** Pair-wise correlation values for the mean annual cycle averaged over the index area,
2006-2017. N=46 at 8-day resolution; significant values are bold.

|  | *Chloro* | *Salt* |  |
|---|---|---|---|
| Chloro |  |  | Chlorophyll+fluorescence |
| Salt | **-0.62** |  | Salinity at 10 m depth |
| SST | 0.34 | **-0.89** | Sea surface temperature |
| U wind | -0.30 | **0.83** | Zonal wind at 10 m height |
| SLP | -0.29 | **0.91** | Sea level air pressure |
| LHF | -0.15 | **0.74** | Evaporation (latent heat flux) |
| SHF | -0.45 | 0.25 | Sensible heat flux |
| Qs | 0.01 | **-0.71** | Net shortwave radiation |
| $Q_L$ | -0.43 | **0.91** | Net longwave radiation |
| curl | 0.32 | **-0.76** | Vorticity of wind stress |
| wv ht | -0.27 | **0.90** | Wave height (sig.) |
| wv dir | -0.30 | **0.63** | Wave direction |
| wv per | -0.21 | **0.70** | Wave period |
| U 10 cur | -0.56 | 0.59 | Zonal current at 10 m depth |
| V 10 cur | -0.55 | **0.79** | Meridional current at 10 m |
| W 30 | 0.36 | -0.55 | Vertical motion at 30 m depth |






**Table 2** Pair-wise correlation of annual fish catch in the outer domain (cf. Fig 8) and Jan-Jun
marine data averaged over the index area, 1981-2015. N=35 at 1-yr resolution; significant
values are bold. Variables are same as in Table 1, except Temp = sea temp at 10 m depth, U2
/ V2 wind = upper level wind (200 hPa).

|  | Squid | Anchovy | Sardine |
|---|---|---|---|
| Squid |  |  |  |
| Anchovy | -0.01 |  |  |
| Sardine | 0.29 | **0.44** |  |
| SST | 0.21 | 0.18 | 0.14 |
| Temp | -0.07 | **0.44** | 0.26 |
| Salt | 0.13 | 0.31 | 0.30 |
| SLP | 0.15 | -0.17 | -0.01 |
| U wind | -0.14 | 0.29 | 0.07 |
| V wind | -0.30 | 0.22 | -0.13 |
| U cur | **-0.54** | -0.09 | -0.06 |
| V cur | **-0.58** | 0.15 | -0.01 |
| LHF | **-0.46** | -0.03 | -0.29 |
| SHF | -0.35 | -0.09 | -0.35 |
| Qs | -0.08 | -0.20 | -0.05 |
| $Q_L$ | -0.28 | -0.08 | -0.19 |
| curl | 0.03 | -0.29 | -0.20 |
| U 2 | -0.23 | 0.31 | 0.18 |
| V 2 | -0.04 | -0.38 | -0.37 |
| rain | 0.22 | 0.07 | 0.21 |


**Figures**

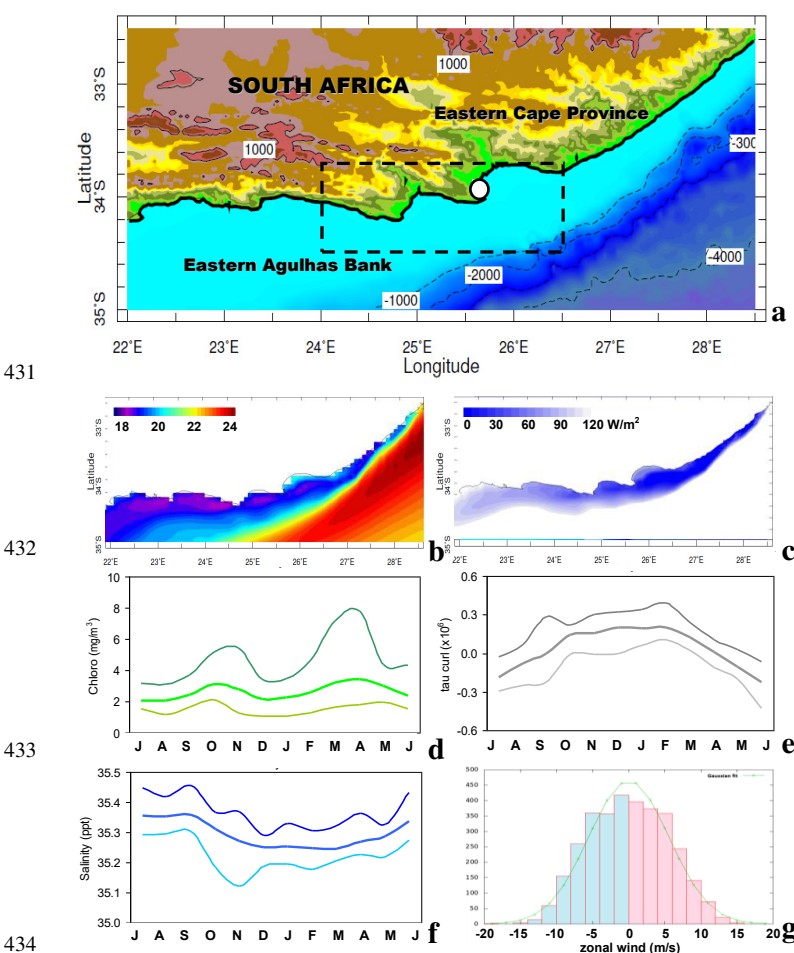

Figure 1 (a) Topography of the outer study domain (and dashed index area), shelf bathymetry
and Port Elizabeth (dot). Mean maps 2006-2017 of HYCOM reanalysis (b) SST (°C) and (c)
net heat flux (shaded < 120 W/m$^2$). Index-area mean annual cycle of: (d) chlorophyll
fluorescence, (e) wind stress curl, and (f) salinity; with upper / lower 2.5 percentiles, summer
is unified. (g) Histogram of index-area daily zonal wind, (blue–from east / red–from west).





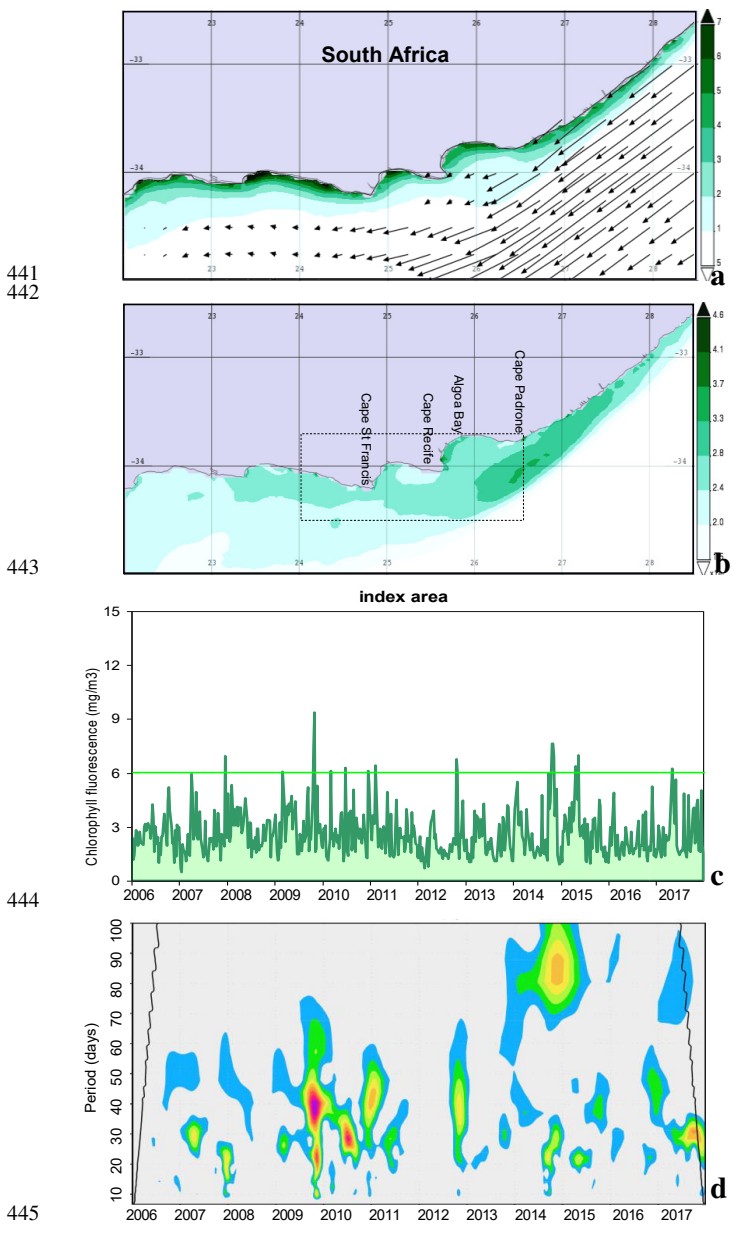

Figure 2 MODIS 2006-2017 mean: (a) chlorophyll concentration (mg/m$^3$) with SODA3 mean

near-surface currents, (b) fluorescence line height (W/m$^2$) and index-area. (c) Time series of

8-day index-area chlorophyll fluorescence; composite cases > 6 mg/m$^3$. (d) Wavelet spectra

energy of the time series, shaded from 90% to 99% confidence.




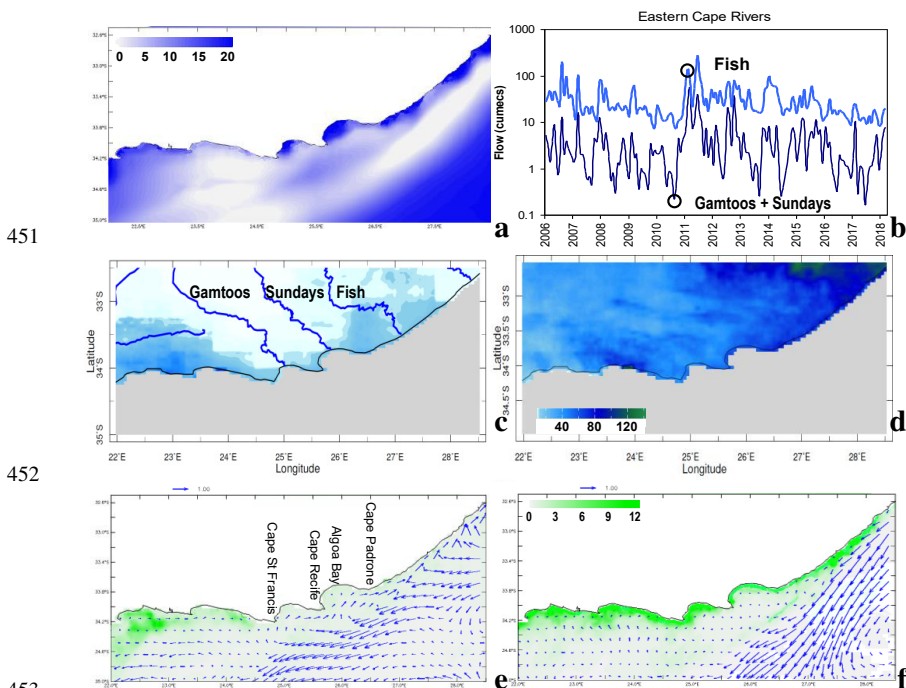




Figure 3 (a) Mean 2006-2017 HYCOM water flux into ocean (mm/day). (b) Observed

monthly river discharge at the coast, with dry spell 1-10 Aug 2010 and wet spell 1-10 Mar

2011 (circled). (c,d) CHIRPS coastal rainfall (cumulative, mm), (e,f) MODIS chlorophyll

(shading) and HYCOM near-surface current vectors during the dry spell (left) and wet spell.

(c) illustrates the main rivers.





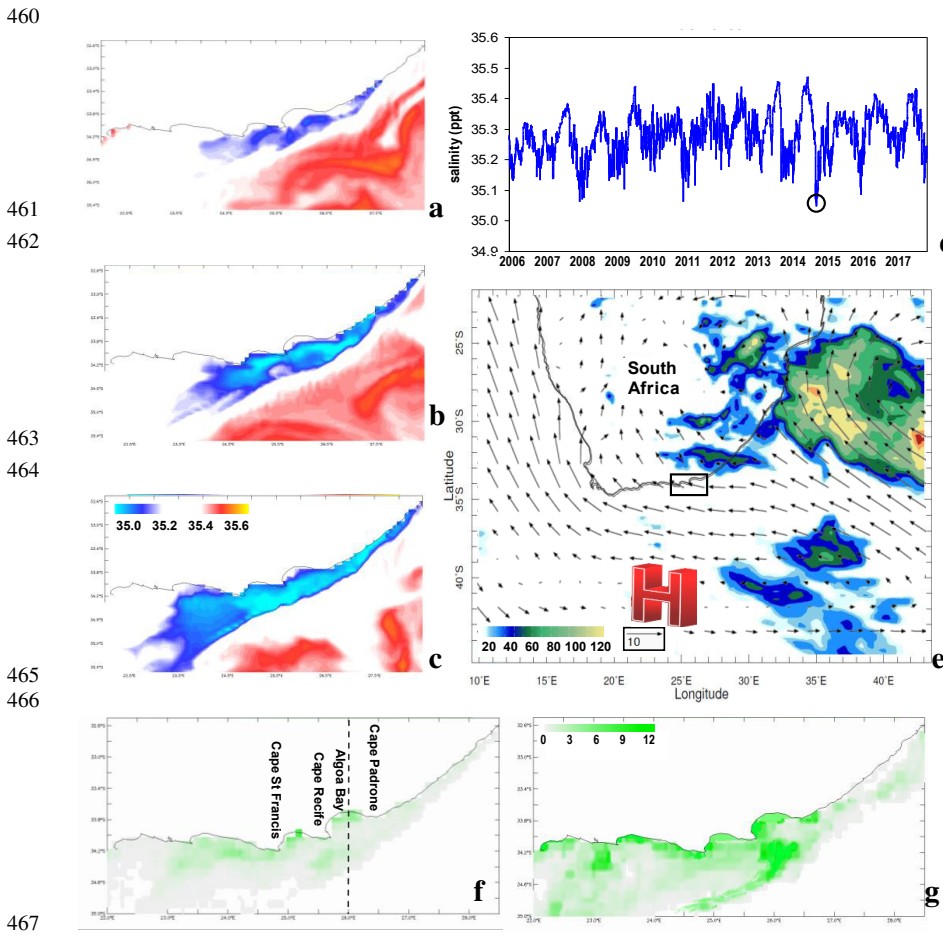



Figure 4 Salinity maps (a) 15-20 Oct, (b) 20-25 Oct, (c) 25-30 Oct 2014, when the
cumulative Fish River discharge exceeded $10^{10}$ m$^3$. (d) Daily record of index-area 10 m
salinity, with case study circled. (e) Large-scale 15-25 Oct 2014 cumulative CMORPH
rainfall (shaded, mm) and low level winds (vector). MODIS chlorophyll: (f) 8-23 Oct, and (g)
24 Oct - 8 Nov 2014. Dashed line in (f) is the section in Fig 5.




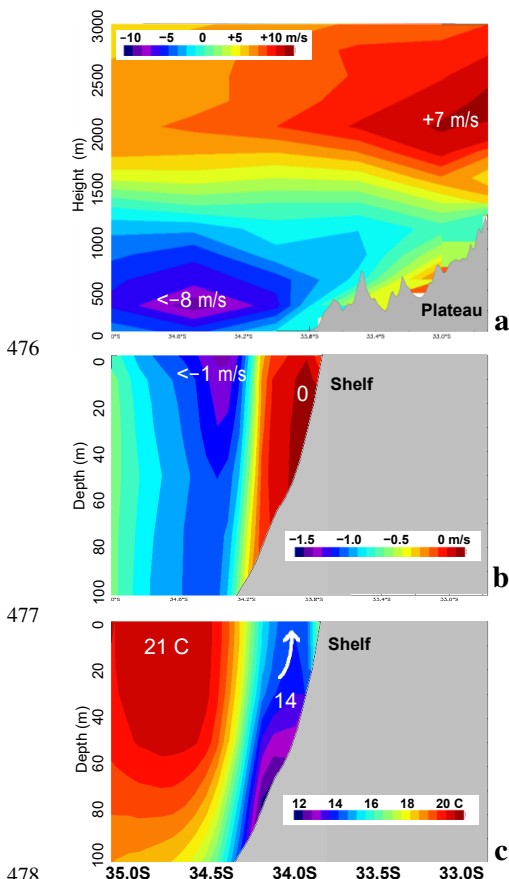




Fig 5 Vertical N-S sections on 26E averaged 15 Oct - 5 Nov 2014 of: (a) MERRA2 zonal
wind, (b) HYCOM zonal current, and (c) sea temperature.


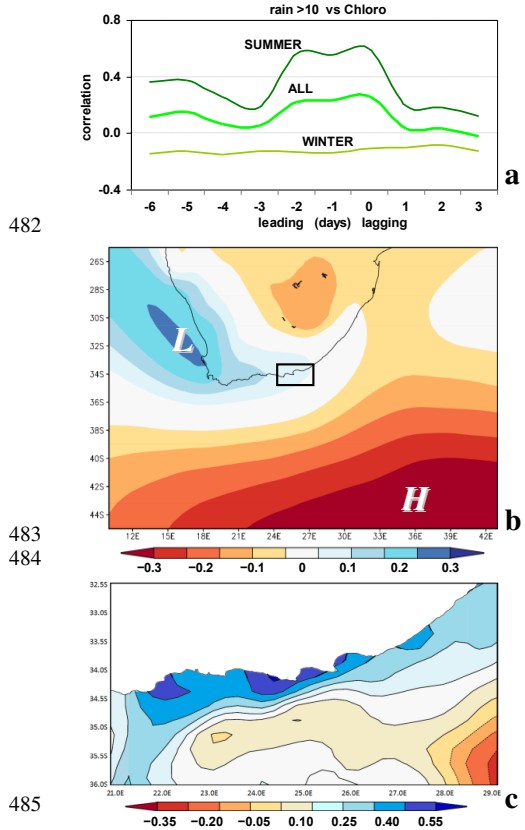




Figure 6 (a) Lag-correlation between index-area coastal rainfall > 10 mm/day and

chlorophyll fluorescence N=102, for summer/ all/ winter season. (b) Point-to-field correlation

of salinity index (cf. Fig 4d) and summer-season field at 3-day lead, Oct-Mar 2006-2017 sea-

level air pressure, icons given w.r.t. lower salinity. (c) Same point-to-field correlation except

with sea-surface height anomalies, shaded w.r.t. lower salinity (eg. low-inshore).






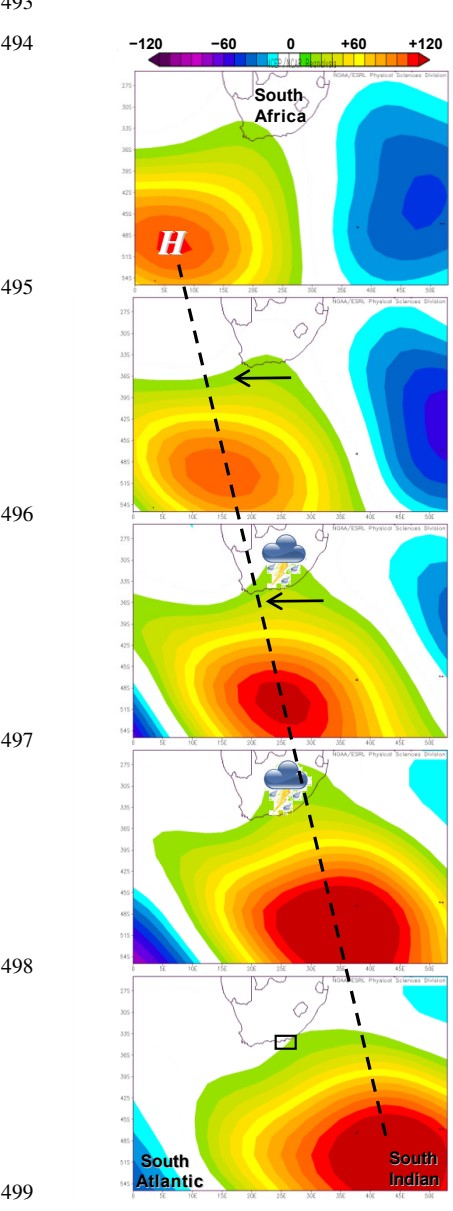






Figure 7 Composite 500 hPa geopotential height anomalies (shaded) for the top 10
chlorophyll fluorescence events, (top-down) from -4 days (before) to zero (colour max), with
dashed line following the mid-latitude ridge. Composite anomaly easterly winds (>10 m/s)
and heavy rainfall (>50 mm) are represented by arrow and icon.



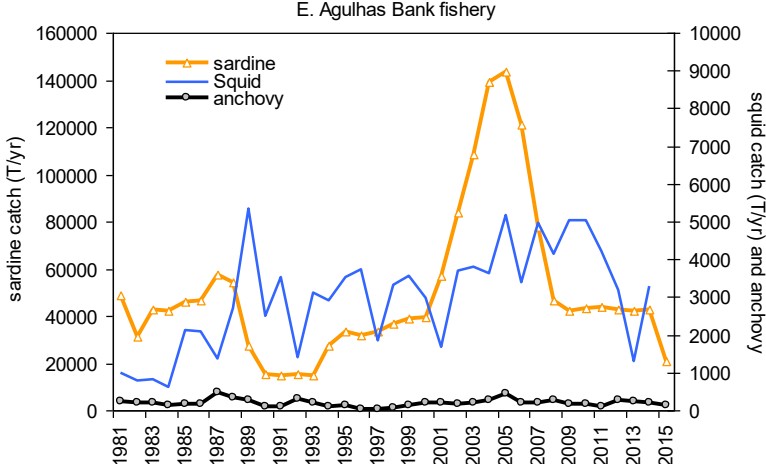


Figure 8 Time series of annual catch for leading fisheries in the eastern Agulhas Bank (outer
domain, cf. Fig 1a,2b); the basis for results in Table 2.





## **Appendix**

Acronyms, dataset and horizontal resolution.

|         | Name                                                        | Horiz. Resolution |
|---------|-------------------------------------------------------------|-------------------|
| CHIRPS  | Climate Hazards InfraRed Precipitation with Station v2      | 5 km              |
| CMORPH  | CPC Morphed polar- and geostationary- satellite rainfall    | 25 km             |
| HYCOM   | Hybrid Coordinate Ocean Model reanalysis from the US Navy   | 8 km              |
| MERRA   | Modern Era Reanalysis for Research and Applications v2       | 50 km             |
| MODIS   | Moderate-imaging Infrared Spectrometer (colour)             | 4 km              |
| SADW    | S.A. Dept of Water Hydrology Service                         | Gauge             |
| SODA    | Simple Ocean Data Assimilation v3 reanalysis                | 25 km             |
| Wave-watch | Wave Analysis Model hindcast v3 coupled with GFS weather model | 50 km          |




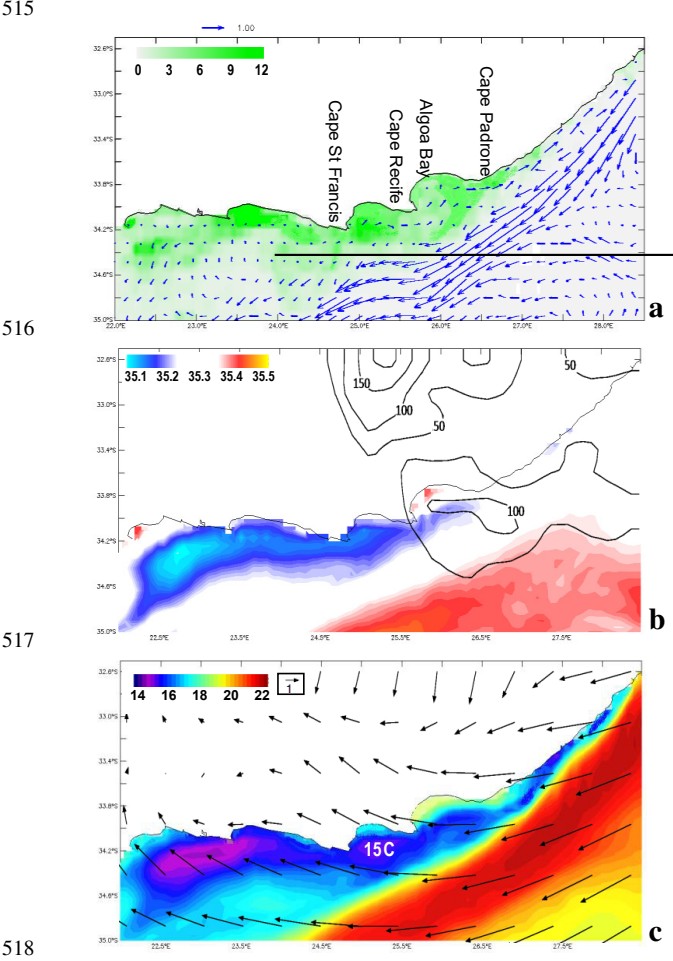


Figure A1  Case study productive event 24 Oct - 8 Nov 2009: (a) MODIS chlorophyll
(mg/m$^3$) and HYCOM currents at 10 m depth (vector), (b) HYCOM salinity at 10 m depth
and cumulative CMORPH rainfall > 50 mm (contour), (c) SST map and MERRA2 surface
winds (vector). Easterly flow, inshore upwelling and high rainfall characterize this event.
Thin west-east line in (a) refers to 34.5S section for Fig A2 below.



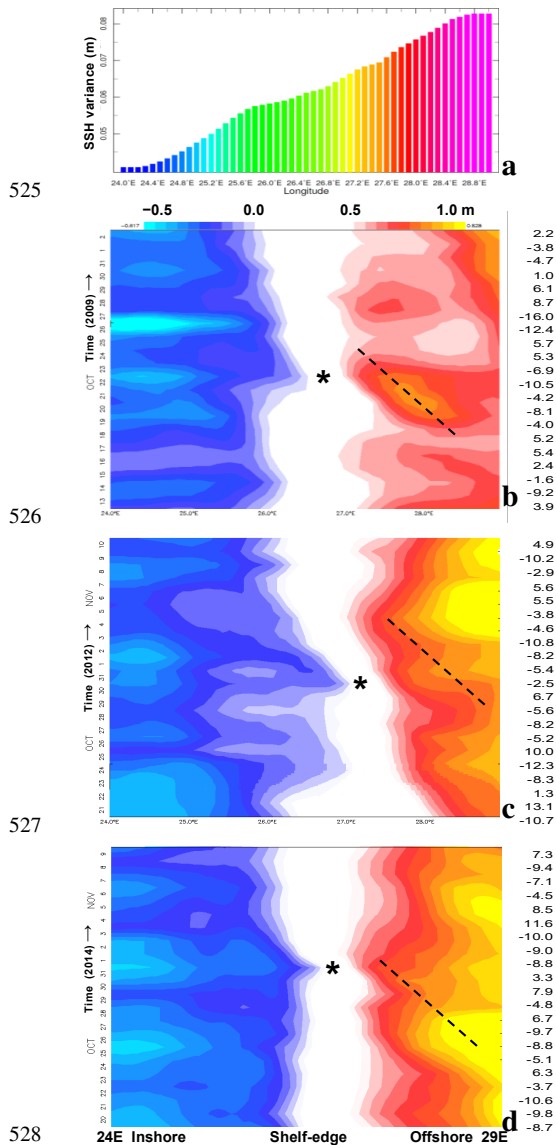





Figure A2  (a) Variance (RMS) of sea surface height anomalies on 34.5S per 0.1° longitude,
2006-2017. Hovmoller plots of HYCOM daily sea surface height on 34.5S over 20 days
during three early summer chlorophyll fluorescence events: (b) 2009, (c) 2012, (d) 2014.
Asterisk highlights steepest SSH gradient; dashed lines highlight ~0.2 m/s westward
movement of anticyclonic warm rings in the offshore zone. Numbers in the right column are
index-area daily zonal winds (m/s, -U from east).