# Peer review of "Environmental controls on marine productivity"

_Ocean Science, 2019_

## Referee Comment (RC1) · Anonymous Referee #1 · 27 Aug 2019

The author acknowledges that this work should be regarded as an explorative investigation of a number of variables available from high resolution re-analysis. Indeed, it seems at times that as many different parameters as possible are thrown into the mix to see where the best correlations could be found. There are results which support the generally held view of the relationship between winds, upwelling and the influence of the Agulhas Current in shelf-edge upwelling. However, the introduction of salinity as an indicator is a novel approach, and the values obtained allow a correspondence to rainfall and river run-off to be made. Nonetheless, the catch statistics in Table 2 show a low correlation and it may be that averaging marine data over a year does not provide the necessary detail. Moreover, averaging parameters over such a wide, dynamic and variable region may lead to suppression of localised and important pro-

cesses which could markedly affect the resultant conditions. Overall, I recommend that the paper be published, on the understanding that some of the results are speculative. However, there are a number of details which require additional clarification; these are listed below.  ć Figure 1(a): Bathymetry less than 1000m is important, and should be shown.  ć It is not clear why the three rivers, Fish, Sundays and Gamtoos were chosen, since there are a number of other large rivers in the area, e.g. Kariega, Bushmans, Swartkops, Kromme etc. Rivers should be shown in Figure 1 and not left to Figure 3.  ć Intense stratification occurs over the wider Agulhas Bank in summer due to insolation and milder winds, while in winter a loss of heat from the surface layers combined with stronger westerly winds causes vertically well-mixed conditions (Schumann and Beekman, Trans Roy Soc SA, 1984, 191-203). Presumably the sensible heat flux of Figure 1(c) is outgoing, supporting the SST of Figure 1(b). This is one of the cases where the seasonal differences are substantial, and are not evident in such averaged portrayals.  ć The south and south-eastern sections of South Africa are not summer rainfall areas, but rather fall between the winter rainfall of the Western Cape and eastern South Africa.  ć Identify parameters $\zeta$ and $\tau$ in line 191  ć Table 1 – why is salt/chloro correlation shown (0.62) and not chloro/salt?  ć The importance of waves is not clear, in particular with the high correlations shown in Table 1.  ć It is not clear where the data used to determine the correlations in Table 2 was obtained for the period 1981 to 2015, in particular values such as sea temperature and currents at 10m.

---

## Referee Comment (RC2) · Anonymous Referee #2 · 17 Sep 2019

General Comments

This exploratory study uses reanalysis products to investigate the ocean and its atmospheric drivers off the Eastern Cape province of South Africa. The influence of these environmental variables on productivity and fisheries are further investigated. The study is novel for the region, in that it uses freely available reanalysis products instead of the traditional in situ measurements or standard regional numerical models. Some in situ measurements made in the study area are assimilated into the reanalysis products, but not many. Apart from remote sensing products, the region is not widely covered in terms of in situ measurements. This makes the use of reanalysis products more useful and in future they will most likely become a necessary component of any large scale study in the region. This study shows the way for local researchers to use a

new tool to investigate atmospheric-ocean interactions in an area short on large-scale in situ measurements.

The main suggestion of the study, as seen by the reviewer, is that the mid-Atlantic high pressure ridge underpins marine productivity in the area. It is this high pressure ridge moving anti-clockwise around southern Africa that is responsible for the easterly winds that drive coastal upwelling and hence productivity. Apart from wind, and rain, the high pressure is also responsible for the sea surface height gradient that contributes towards shelf and coastal ocean processes. The Agulhas Current also plays a big role through pressure gradients, nearshore currents and both cold and warm water intrusions. These all play a role in determining the marine productivity of the area.

The interpretation of the chlorophyll distribution between dry and wet conditions needs further clarification. Chlorophyll and currents change drastically over a period of days, while the change from dry to wet conditions happened over months. So, the author is comparing processes that occur over different time scales, and this could lead to an incorrect interpretation.

The movement of low salinity water south-westward from the Fish River needs to be checked. Can fresh water from the Fish River actually move that far across the shelf in that amount of time, and without dispersing? What about the counter coastal current, towards the north, that could hamper the south-westward movement of water? To the reviewer, the lower salinity structure shown in Fig 4 looks very similar to a structure that is observed after extensive, strong coastal upwelling when high chlorophyll is observed along the open-ocean side of joined upwelling plumes that originated at the capes.

A few improvements could be made to the paper. Figures are small, making them difficult to read, but the figures could be fine in the published paper so maybe it is not important. References are missing. There could be more references in the discussion, and even the results, if references in the results are acceptable.

I recommend that the paper be published.

[Figure]

Specific Comments

38: Incorrect. Easterly wind increase during summer and decrease during winter while westerly winds are more or less constant throughout the year.

145:Fig 2a shows chlorophyll as lower around the capes as compared to the coastal band (band same distance offshore), not higher.

145: Fig 2b shows fluorescence as lower around the capes as compared the shelf.

148-9: The plume off Cape Recife is associated with wind-driven upwelling, not Agulhas Driven upwelling.

163-168: The time difference from dry conditions (Aug 2010) to wet condition (Mar 2011) had a span of approx. 7 months, as you state. Both coastal chlorophyll and currents (shelf and the Agulhas Current) change over periods of days or several days, or maybe even weeks in the case of the Current. To try correlate events fluctuating (or happening) at vastly different timescales, as shown by the 2 matching figures (Fig3e and 3f) as compared to rainfall, can lead to the wrong interpretation. The chlorophyll and currents could, and probably did, change many times over that 7 month period and it may just be that the day snapshots that you chose were biased towards your reasoning.

170: Should use "high/er salinity" and not "salty".

170: You should place the Fish River on Fig 4. St Francis Bay, Algoa Bay and the coastal zone north to what appears to be the position of the Fish River had higher salinity than the shelf waters (∼35.3 compared to ∼35.1). Don't you find that strange?

169-177: Anyway, if the Fish River position were known, then, yes, it would appear from your plots that lower salinity water filled the mid-shelf region with its northern extremity around the Fish River. What concerns me is this: would lower salinity water move westward and fill the whole mid-shelf area (distance of ∼300km) over a period of 5-6 days (Fig4a to Fig 4b) without mixing and dispersing, and could it even do that from

theory? If it were possible, then, yes, your reasoning would be considered sound.

169-177: Yes, your Fig 4e does show a net easterly wind from 15-25 Oct 2014. From wind records, there were moderate to fresh breezes from 17-23 Oct, the rest of the days being westerly winds. So, 6 days of easterly winds (if we forget the westerly winds) to disperse a surface plume ~300 km. Which gives a downwind velocity of ~0.58 m/s. From local ADCP records, the downwind surface velocity of a particle moving with the water at that speed is not impossible. However, that low salinity structure looks very similar to a high surface chlorophyll structure that occurs after extensive, strong coastal upwelling. Which makes me wonder?

179: Your text says 17 Oct – 8 Nov whereas Fig 5 says 15 Oct - 5 Nov. Otherwise, I agree with what is presented in paragraph 178-188.

189-196: This paragraph reads more like a discussion. Upwelling along the inshore edge of the Agulhas Current, which moves along the bottom over the shelf in the direction of the coastline "primes" the capes for wind-driven upwelling. So, the upwelling response to easterly winds in almost immediate at the shoreline, whereas downwelling during westerly winds just "restores' the ocean back to normal.

197-202: I'm not sure about this. Maybe more explanation is needed to make it relevant.

210: Yes, I imagine that high offshore sea pressure could contribute to enhanced westward flow near at the coast and hence contribute positively towards upwelling. Actually, just that has been found from in situ data during Natal Pulses by Goschen et al (2015).

217: Agreed, large scale weather systems moving eastward over southern Africa produce easterly winds (among other phenomena), which are upwelling favourable.

Technical Errors

220 and 222: Should that not be "chlorophyll and fluorescence"? Are they not two different ocean variables?

---

## Author Comment (AC1) · 1 Oct 2019

Referee #1 Interactive comment on Ocean Sci. Discuss., https://doi.org/10.5194/os-2019-55, 2019.

AUTHOR REPLIES IN CAPITALS The author acknowledges that this work should be regarded as an explorative investigation of a number of variables available from high resolution re-analysis. Indeed, it seems at times that as many different parameters as possible are thrown into the mix to see where the best correlations could be found. There are results which support the generally held view of the relationship between winds, upwelling and the influence of the Agulhas Current in shelf-edge upwelling. However, the introduction of salinity as an indicator is a novel approach,

and the values obtained allow a correspondence to rainfall and river run-off to be made. Nonetheless, the catch statistics in Table 2 show a low correlation and it may be that averaging marine data over a year does not provide the necessary detail. Moreover, averaging parameters over such a wide, dynamic and variable region may lead to suppression of localised and important processes which could markedly affect the resultant conditions. Overall, I recommend that the paper be published, on the understanding that some of the results are speculative. However, there are a number of details which require additional clarification; these are listed below. Figure 1(a): Bathymetry less than 1000m is important, and should be shown. THE -100 AND -200 ISOBATHS WERE ADDED. It is not clear why the three rivers, Fish, Sundays and Gamtoos were chosen, since there are a number of other large rivers in the area, e.g. Kariega, Bushmans, Swartkops, Kromme etc. Rivers should be shown in Figure 1 and not left to Figure 3. THOSE 3 RIVERS HAVE THE GREATEST DISCHARGE AND APPEAR TO AFFECT THE SHELF SALINITY MORE THAN OTHERS, HENCE THEIR DATA WAS ANALYZED. THE AUTHOR FEELS THAT THE RIVERS WILL BE 'LOST' IN THE COMPLEX TOPOGRAPHY OF FIG 1, AND SO PREFERS TO KEEP RIVERS IN FIG 3 WITH WATER FLUX Intense stratification occurs over the wider Agulhas Bank in summer due to insolation and milder winds, while in winter a loss of heat from the surface layers combined with stronger westerly winds causes vertically well-mixed conditions (Schumann and Beekman, Trans Roy Soc SA, 1984, 191-203). YES, THAT WILL BE ADDED. Presumably the sensible heat flux of Figure 1(c) is outgoing, supporting the SST of Figure 1(b). This is one of the cases where the seasonal differences are substantial, and are not evident in such averaged portrayals. AGREED - HENCE, THE AUTHOR COVERS THE SEASONAL CYCLE IN FIG 1D,E,F. The south and south-eastern sections of South Africa are not summer rainfall areas, but rather fall between the winter rainfall of the Western Cape and eastern South Africa. AGREED. Identify parameters $\zeta$ and $\tau$ in line 191 Table 1 – why is salt/chloro correlation shown (0.62) and not chloro/salt? OK, PARAMETERS IDENTIFIED, THE CORRELATION IS LISTED IN TABLE 1 AND DUPLICATE RESULTS ARE OMITTED.
The importance of waves is not clear, in particular with the high correlations shown in Table 1. WAVES DERIVE FROM WINTER STORMS AND BUT RAINFALL AND RUN-OFF PEAK IN SUMMER. THAT IS NOW MENTIONED. It is not clear where the data used to determine the correlations in Table 2 was obtained for the period 1981 to 2015, in particular values such as sea temperature and currents at 10m. THAT IS MENTIONED IN DATA SECTION, USING SODA3 REANALYSIS, EXCEPT FOR UPPER WINDS, SST AND RAIN. THOSE ARE IDENTIFIED IN TABLE 2 CAPTION.

Please also note the supplement to this comment:
https://www.ocean-sci-discuss.net/os-2019-55/os-2019-55-AC1-supplement.zip

---

## Author Comment (AC2) · 1 Oct 2019

Reviewer 2 comment: AUTHOR REPLIES IN CAPITALS

General Comments: This exploratory study uses reanalysis products to investigate the ocean and its atmospheric drivers off the Eastern Cape province of South Africa. The influence of these environmental variables on productivity and fisheries are further investigated. The study is novel for the region, in that it uses freely available reanalysis products instead of the traditional in situ measurements or standard regional numerical models. Some in situ measurements made in the study area are assimilated into the reanalysis products, but not many. Apart from remote sensing products, the region is

not widely covered in terms of in situ measurements. This makes the use of reanalysis products more useful and in future they will most likely become a necessary component of any large scale study in the region. This study shows the way for local researchers to use a new tool to investigate atmospheric-ocean interactions in an area short on large-scale in situ measurements.

The main suggestion of the study, as seen by the reviewer, is that the mid-Atlantic high pressure ridge underpins marine productivity in the area. It is this high pressure ridge moving anti-clockwise around southern Africa that is responsible for the easterly winds that drive coastal upwelling and hence productivity. Apart from wind, and rain, the high pressure is also responsible for the sea surface height gradient that contributes towards shelf and coastal ocean processes. The Agulhas Current also plays a big role through pressure gradients, nearshore currents and both cold and warm water intrusions. These all play a role in determining the marine productivity of the area. SOME OF THIS INSIGHT IS REITERATED IN THE CONCLUSIONS.

The interpretation of the chlorophyll distribution between dry and wet conditions needs further clarification. Chlorophyll and currents change drastically over a period of days, while the change from dry to wet conditions happened over months. So, the author is comparing processes that occur over different time scales, and this could lead to an incorrect interpretation. YES, THAT VARIABILITY IS A KEY POINT AND COVERED IN FIG 2C. IT IS AGREED THAT FLUCTUATIONS ARE CHAOTIC AS EVIDENCED BY THE SPOTTY APPEARANCE OF THE WAVELET SPECTRAL ANALYSIS IN FIG 2D. THAT IS NOW MENTIONED IN THE SUMMARY.

The movement of low salinity water south-westward from the Fish River needs to be checked. Can fresh water from the Fish River actually move that far across the shelf in that amount of time, and without dispersing? RECALL THAT HYCOM IS SUP-PORTED BY NCODA, SO ANY CHANGES IN SALINITY ARE NOT SIMULATED, BUT OBSERVED VIA DATA ASSIMILATION, EG. SATELLITE RAINFALL, SOIL MOIS-TURE OVER THE COASTAL PLAINS, AND SALINITY MEASUREMENTS OVER THE

[Figure]

SHELF – SUBSEQUENTLY ADVECTED ACCORDING TO SATELLITE ALTIMETER ESTIMATED NEAR-SURFACE CURRENTS. What about the counter coastal current, towards the north, that could hamper the south-westward movement of water? To the reviewer, the lower salinity structure shown in Fig 4 looks very similar to a structure that is observed after extensive, strong coastal upwelling when high chlorophyll is observed along the open-ocean side of joined upwelling plumes that originated at the capes. THIS STATEMENT IS USEFUL AND WAS ADDED IN THE RESULTS SECTION.

THE LOCAL SST-SALT CORRELATION WAS NEGATIVE OVER THE ANNUAL CYCLE (TABLE 1, R = -.89) SO UPWELLING SHOULD BE OVERLAIN WITH THE RIVER PLUMES TO ENHANCE PRODUCTIVITY. ANOTHER GOOD POINT FROM THE REVIEWER, WHICH IS ADDED TO THE CONCLUSIONS.

A few improvements could be made to the paper. Figures are small, making them difficult to read, but the figures could be fine in the published paper so maybe it is not important. References are missing. There could be more references in the discussion, and even the results, if references in the results are acceptable. I recommend that the paper be published. A FEW REFERENCES WERE ADDED.

Specific Comments 38: Incorrect. Easterly wind increase during summer and decrease during winter while westerly winds are more or less constant throughout the year. YES, IT WAS ALREADY SAID THAT EASTERLIES INCREASE IN SUMMER. 145:Fig 2a shows chlorophyll as lower around the capes as compared to the coastal band (band same distance offshore), not higher. YES, IT IS DOWNSTREAM FROM THE CAPES WHERE PRODUCTIVITY IS HIGHER, THAT IS CHANGED. 145: Fig 2b shows fluorescence as lower around the capes as compared the shelf. AS ABOVE. 148-9: The plume off Cape Recife is associated with wind-driven upwelling, not Agulhas Driven upwelling. YES, CAPE PADRONE HAS THE STRONGEST CURRENT INDUCED SHEAR AND SHELF-EDGE UPWELLING. 163-168: The time difference from dry conditions (Aug 2010) to wet condition (Mar 2011) had a span of approx. 7
months, as you state. Both coastal chlorophyll and currents (shelf and the Agulhas Current) change over periods of days or several days, or maybe even weeks in the case of the Current. To try correlate events fluctuating (or happening) at vastly different timescales, as shown by the 2 matching figures (Fig3e and 3f) as compared to rainfall, can lead to the wrong interpretation. The chlorophyll and currents could, and probably did, change many times over that 7 month period and it may just be that the day snapshots that you chose were biased towards your reasoning. THE RIVER DISCHARGE TIME SERIES (FIG 3B) SHOWS A LENGTHY DRY SPELL. 'SNAPSHOTS' COVER 5-DAY INTERVALS IN A SEQUENCE. 170: Should use "high/er salinity" and not "salty". YES, CHANGED. 170: You should place the Fish River on Fig 4. St Francis Bay, Algoa Bay and the coastal zone north to what appears to be the position of the Fish River had higher salinity than the shelf waters (âĹij35.3 compared to âĹij35.1). Don't you find that strange? THE RIVERS ARE PLACED ON FIG 3, AN ARROW WAS ADDED TO FIG 4, TO SHOW THE FISH RIVER MOUTH. 169-177: Anyway, if the Fish River position were known, then, yes, it would appear from your plots that lower salinity water filled the mid-shelf region with its northern extremity around the Fish River. What concerns me is this: would lower salinity water move westward and fill the whole mid-shelf area (distance of âĹij300km) over a period of 5-6 days (Fig4a to Fig 4b) without mixing and dispersing, and could it even do that from theory? If it were possible, then, yes, your reasoning would be considered sound. IN ADDITION TO THE RIVER OUTFLOW THERE WAS WIDESPREAD RAINFALL AS SHOWN IN FIG 4E. THIS IS NOW EMPHASIZED. 169-177: Yes, your Fig 4e does show a net easterly wind from 15-25 Oct 2014. From wind records, there were moderate to fresh breezes from 17-23 Oct, the rest of the days being westerly winds. So, 6 days of easterly winds (if we forget the westerly winds) to disperse a surface plume âĹij300 km. Which gives a downwind velocity of âĹij0.58 m/s. From local ADCP records, the downwind surface velocity of a particle moving with the water at that speed is not impossible. However, that low salinity structure looks very similar to a high surface chlorophyll structure that occurs after extensive, strong coastal upwelling. Which makes me wonder? THE

EASTERLY WINDS CAUSE SHELF WATERS TO JOIN WITH THE AGULHAS AND ADVECT RIVER PLUMES DOWNSTREAM. THE HYCOM MODEL CAPTURES THIS PROCESS VIA SATELLITE DATA ASSIMILATION. WIND VARIABILITY DURING MAJOR UPWELLING EVENTS IS COVERED IN APPENDIX A2. 179: Your text says 17 Oct – 8 Nov whereas Fig 5 says 15 Oct - 5 Nov. Otherwise, I agree with what is presented in paragraph 178-188. OK TEXT WAS CHANGED, TO SAY 'LATE OCT 2014'. 189-196: This paragraph reads more like a discussion. Upwelling along the inshore edge of the Agulhas Current, which moves along the bottom over the shelf in the direction of the coastline "primes" the capes for wind-driven upwelling. THESE PROCESSES OPERATE AT SLIGHTLY DIFFERENT TIME AND SPACE SCALES - SHELF-EDGE UPWELLING IS ADVECTED INTO THE AREA, WHILE THE COASTAL UPWELLING IS LOCALIZED. So, the upwelling response to easterly winds in almost immediate at the shoreline, whereas downwelling during westerly winds just "restores' the ocean back to normal. 197-202: I'm not sure about this. Maybe more explanation is needed to make it relevant. OK. 210: Yes, I imagine that high offshore sea pressure could contribute to enhanced westward flow near the coast and hence contribute positively towards upwelling. Actually, just that has been found from in situ data during Natal Pulses by Goschen et al (2015). 217: Agreed, large scale weather systems moving eastward over southern Africa produce easterly winds (among other phenomena), which are upwelling favourable. Technical Errors 220 and 222: Should that not be "chlorophyll and fluorescence"? Are they not two different ocean variables? THE PRODUCTIVITY INDEX USES AN AVERAGE OF BOTH, SO IT IS REFERRED TO AS CHLOROPHYLL FLUORESCENCE.

Please also note the supplement to this comment:
https://www.ocean-sci-discuss.net/os-2019-55/os-2019-55-AC2-supplement.zip